# Potentially Inappropriate Prescribing and Potential Prescribing Omissions in 82,935 Older Hospitalised Adults: Association with Hospital Readmission and Mortality within Six Months

**DOI:** 10.3390/geriatrics5020037

**Published:** 2020-06-12

**Authors:** Roger E. Thomas, Leonard T. Nguyen, Dave Jackson, Christopher Naugler

**Affiliations:** 1Department of Family Medicine, Cumming School of Medicine, University of Calgary, Calgary, AB T2L 2K8, Canada; ctnaugle@ucalgary.ca; 2Data Analyst, Alberta Precision Laboratories, Alberta Health Services, Calgary, AB T2N 4N1, Canada; leonard.nguyen@aplabs.ca; 3Data Analyst, Airdrie & Area Health Cooperative, Airdrie, AB T4B OR6, Canada; davejackson41@gmail.com; 4Department of Pathology & Laboratory Medicine, Cumming School of Medicine, University of Calgary, Calgary, AB T2N 4N1, Canada; 5Department of Community Health Sciences, Cumming School of Medicine, University of Calgary, Calgary, AB T2N 4N1, Canada; 6Undergraduate Medical Education, Cumming School of Medicine, University of Calgary, Calgary, AB T2N 4N1, Canada

**Keywords:** polypharmacy, potentially inappropriate medications, potential prescribing omissions, hospital readmission, mortality, STOPP/START criteria, American Geriatric Society criteria

## Abstract

Polypharmacy with “potentially inappropriate medications” (PIMs) and “potential prescribing omissions” (PPOs) are frequent among those 65 and older. We assessed PIMs and PPOs in a retrospective study of 82,935 patients ≥ 65 during their first admission in the period March 2013 through February 2018 to the four acute-care Calgary hospitals. We used the American Geriatric Society (AGS) and STOPP/START criteria to assess PIMs and PPOs. We computed odds ratios (ORs) for key outcomes of concern to patients, their families, and physicians, namely readmission and/or mortality within six months of discharge, and controlled for age, sex, numbers of medications, PIMs, and PPOs. For readmission, the adjusted OR for number of medications was 1.09 (1.09–1.09), for AGS PIMs 1.14 (1.13–1.14), for STOPP PIMs 1.15 (1.14–1.15), for START PPOs 1.04 (1.02–1.06), and for START PPOs correctly prescribed 1.16 (1.14–1.17). For mortality within 6 months of discharge, the adjusted OR for the number of medications was 1.02 (1.01–1.02), for STOPP PIMs 1.07 (1.06–1.08), for AGS PIMs 1.11 (1.10–1.12), for START PPOs 1.31 (1.27–1.34), and for START PPOs correctly prescribed 0.97 (0.94–0.99). Algorithm rule mining identified an 8.772 higher likelihood of mortality with the combination of STOPP medications of duplicate drugs from the same class, neuroleptics, and strong opioids compared to a random relationship, and a 2.358 higher likelihood of readmission for this same set of medications. Detailed discussions between patients, physicians, and pharmacists are needed to improve these outcomes.

## 1. Introduction

### 1.1. Background and Rationale

Family physicians care for patients with multimorbidity in up to 80% of their consultations, and for geriatricians in nearly all of their consultations. These patients take multiple medications and a systematic review concluded that to avoid interactions between the vast potential heterogeneity of illnesses and polypharmacy, ascertaining patient goals and preferences, shared goal setting and decision making, and individualised care with excellent continuity to optimise benefits over harms, are required [1].

A systematic review identified that the most frequent definition of polypharmacy in adults was five or more medications daily, but only 6% of the 138 definitions assessed comorbidities and the appropriateness or safety of therapy to define polypharmacy in a wider context [2]. However, some patients with multiple complex diseases may appropriately need polypharmacy. An important concern is whether polypharmacy is associated with mortality, and a systematic review of 47 studies (28 were assessed as high quality and 19 as medium on the Newcastle-Ottawa scale) found that the odds ratio of death was 1.24 (95%CI 1.10, 1.39) for one to four medications, 1.31 (1.17, 1.47) for five medications, 1.59 (1.36, 1.87) for six to nine, and 1.96 (1.42, 2.71) for 10 or more medications [3].

Another concern is whether polypharmacy is associated with preventable adverse drug reactions (ADRs). A systematic review of systematic reviews of in-patients of all ages identified 13 reviews, which assessed 37 primary studies. Thirteen of the studies used prospective reporting (which tends to report higher rates than retrospective or voluntary reporting) and found an incidence of 3.13 (2.87, 3.38) per 100 patients, but due to the inadequate reporting of patients’ ages, rates were not computed by age [4]. A prospective review of 6427 cases of adverse drug reactions recorded in the German Network of Regional Pharmacovigilance Centers in 2000–2008 identified that adverse drug reactions occurred in 3.3% of admissions to internal medicine wards, and that 60% occurred in those 70 and older. Risk factors for those 70 and older were multimorbidity, two to four ADR-causative drugs, and specific drugs (aspirin, phenoprocoumon, insulin, digitoxin, diclofenac, metoprolol, torasemide, glyburides, and spironolactone), but not the total numbers of medications or potentially inappropriate medications (PIMs). The authors recommended individualised medication reviews to identify the most implicated drugs [5]. A review of adverse drug reactions in the US Veterans Affairs database in 2009–2016 identified a 15% rate of adverse drug reactions for those 60–69, 13% for 70–79, 11% for 80–89, and 9% for those 90 and older, and 5% of ADRs were rated as severe [6].

Potentially inappropriate prescribing occurs in many countries. A 2019 systematic review identified 60 cohort and two randomised controlled studies in 32 countries [7]. For thirty studies which used the STOPP/START 2015 [8] criteria (*n* = 1,245,974) the average percentages of patients with one or more “potentially inappropriate medication” (PIM) was 42.8% for 1,242,010 community patients and 51.8% for 3964 hospitalised patients, and for nineteen studies which used the American Geriatric Society 2015 criteria [9], they were 58% for 593,389 community and 55.5% for 2422 hospitalised patients. Only twelve studies measured changes over time, with ten reporting PIM decreases (seven used interventions to reduce polypharmacy and three used standard admissions procedures). Only eight studies assessed correlations between American Geriatric Society (AGS) PIMs, STOPP PIMs, and START “potential prescribing omissions” (PPOs) and rehospitalisation, and they found significant correlations with subsequent hospitalisation [10,11,12,13,14] and the duration of hospital stay [15,16,17]. Although these studies included 1,854,698 patients, they provide limited data about the relationship between PIMs and PPOs for key patient outcomes, such as rehospitalisation or death. Moreover, only 11,061 hospitalised patients (0.6% of 1,854,698) were assessed [7].

### 1.2. Objectives

1. To identify the numbers and identities of AGS and STOPP/START PIMs and PPOs at admission and discharge in a large retrospective database of hospitalised patients aged 65 and older. 2. Identify how many patients received prescriptions to correct PPOs. 3. Identify correlations of age, sex, comorbidities, and individual and groups of medications with the key outcomes of concern to patients, their families, and physicians, namely rehospitalisation and mortality. 4. Assess how hospitalisation (and thus the effects of being assessed by a new medical team) affected the numbers of PIMs and PPOs.

## 2. Materials and Methods

### 2.1. Study Design, Participants, and Setting

The study is a retrospective anonymised database of charts of 82,935 hospitalised patients aged 65 or older admitted to the four acute-care Calgary hospitals (Foothills Medical Centre, Rockyview General Hospital, Peter Lougheed Centre, and South Health Campus) over five years and discharged from 1 March 2013 to 28 February 2018. Their first visit in this period is the focus of analysis for this article (Figure 1). All four hospitals provide full spectrum acute care and Rockyview has urology as an additional specialty. Calgary is a major Canadian city with a population of 1.3 million. Subsequent readmissions will be the subject of a later article.

### 2.2. Variables

The variables of interest are age, sex, admission and discharge diagnoses, comorbidities (listed here in order of frequency: hypertension, diabetes, cardiovascular disease, arthritis, dementia or cognitive decline, atrial fibrillation, falls, fractures, osteoporosis, pneumonia, heart failure, Chronic Obstructive Lung Disease (COPD), urinary tract infection, pain, renal failure, coronary artery disease, myocardial infarction, peptic ulcer, stroke, and chronic renal disease), and medications. Medications were entered as the usual dosage (whether a single tablet or multiple tablets daily constituted a normal prescription). “Potentially inappropriate medications” (PIMs) were assessed using the two most widely used criteria, the AGS list [9] and the Screening Tool of Older People’s Prescriptions (STOPP) [8].

For each admission, diagnoses, co-morbidities, and data on admission and discharge medications permitted the assessment of 78/80 STOPP PIMs and 28/34 START PPOs. We did not have ethics permission or the resources to perform detailed chart reviews on 82,935 patients. STOPP PIMs not assessable with the available data were: drugs prescribed without evidence-based clinical indication (A1) and prescribed beyond recommended duration (A2) and START PPOs were: home continuous oxygen with chronic hypoxaemia (B3), fibre supplement for diverticulosis with constipation, annual influenza (I1) and pneumococcal (I2) vaccines, and vitamin D/calcium supplements for musculoskeletal issues (E2, E3, E5). We also measured whether the PPO medications recommended by the START criteria were actually prescribed. We were able to assess 69 AGS criteria, but not PIMs affecting the renal system, which required laboratory data that were largely unavailable.

We chose as the outcome measures most closely related to patient welfare all-cause rehospitalisation and all-cause death within six months of discharge.

### 2.3. Data Sources and Measurement

Admission and discharge records, laboratory data, and medications were accessed from the Alberta Health Services (AHS) registration database and the Pharmaceutical Information Network (PIN) through the Alberta Health Service’s Data Integration, Management, and Reporting database (DIMER). DIMER statisticians anonymised all patient information and only aggregate data were provided to the researchers and are presented here. Admission medications and dosages were as dispensed to patients before admission, and discharge medications as dispensed during the hospital stay and prescribed to be continued after discharge.

Admission and discharge diagnoses and comorbidities were obtained from admission and discharge summaries. Patients’ illnesses may progress rapidly and their medications may change frequently, and we assessed a decade as too long a period during which to monitor change, so five-year age groups were chosen to identify periods during which adverse change might occur. The STOPP/START and the American Geriatric Society Beers criteria both classify medications by anatomical and then therapeutic classifications (A/T) and their detailed listing of individual medications and groups of medications under their A/T classifications is utilised in Appendix A.

The electronic medical records in the four hospitals during this period did not constrain physicians to enter admission and discharge diagnoses and comorbidities as ICD-9 or ICD-10 codes, and a dictionary was constructed to reduce the multiple ways in which diagnoses were entered to common terms. It was deemed incorrect post facto to construct ICD-10 codes without individual examination of each chart. There was no specific category for ADRs in the hospitals’ electronic medical records; our estimate of confidence in the rate we derived was very low and we chose not to report it rather than report data for which the quality of reporting was unknown.

The data were entered in the statistical package R studio [17,18] and logistic regression was used to compute correlations between age, sex, numbers of medications on admission and discharge, numbers of PIMs and PPOs, and individual and groups of PIMs and PPOs with rehospitalisation or death within six months of discharge. Odds ratios for medications describe the relationship of increases in numbers of medications, PIMs, or PPOs to outcome measures.

The literature review highlighted the need to also identify individual problematic medications. Association rule mining (ARM) [19,20,21,22] was used to identify individual medications and groups of PIM and PPO medications which correlated with rehospitalisation or death within six months of discharge. The arules package’s apriori algorithm [23] was executed in the R statistical package to assess 82,935 visits: each PIM or PPO medication was grouped with other PIM or PPO medications in datasets of medications and compared to the outcomes of rehospitalisation or death within six months of discharge. Each correlation was then compared to the correlation if the datasets and outcomes were independent of each other. The first rule (the support threshold) was set at 0.01 (to avoid generating too many rules with low associations) thus requiring that a PIM and an outcome must occur for at least 1% (829) of the patients. This resulted in 99 rules for AGS PIM sets, 185 rules for STOPP PIM sets, 15 rules for START PPO sets (a lower support threshold of 0.005 was chosen with the smaller number of PPOs), and 76 rules for START medications correctly prescribed for patients. These rules were then ranked according to the degree of lift (indicating the degree to which a PIM set is associated with a compared outcome if the events were completely independent) (Appendix A).

Ethics approval was provided by the Conjoint Health Research Ethics Board, University of Calgary (ID REB15–2163).

From 1 March 2013 to 28 February 2018 the first admission of all patients ≥ 65 years admitted to the four acute-care Calgary hospitals (Foothills Medical Centre, Rockyview General Hospital, Peter Lougheed Centre, and South Health Campus) constituted the sample (*n* = 82,935) (Figure 1). We included elective surgeries, as these patients could also have PIMs/PPOs. These patients had 97,062 readmissions over the five-year period and are the subject of another article. The current study population excludes those admitted for palliative care (whose medications were likely to differ substantially from their usual medications) or who died during this first admission (and thus by definition had no medications subsequent to discharge).

## 3. Results

### 3.1. Sample and Medications at Admission and Discharge

The sample comprised 82,935 patients with a median age of 75 years and 50.5% were females (Table 1).

The most frequent medications are listed in Figure 2a and the most frequent health issues were hypertension (39.6%), diabetes (21.0%), cardiovascular disease (19.7%), arthritis (14.3%), and dementia/cognitive impairment (13.6%) (Figure 2b). IQR = interquartile range; 6M = 6 months.

At admission, patients had a median of four (IQR = 2–7, maximum 28) prescribed medications and nine (IQR = 5–13, maximum 63) at discharge with minimal differences between males and females (Table 1). The most frequent medication class at discharge was statins (prescribed to 44.7% of patients), followed by alpha-1 blockers (43.7%), proton pump inhibitors (42.1%), opioids (39.8%), and beta-blockers (36.9%) (Figure 2a). These percentages correspond to large numbers of patients on statins (37,072), alpha blockers (36,243), proton pump inhibitors (34,916), opioids (32,759) and beta blockers (30,603).

The frequencies of PIMs/PPOs are shown in Table 2 and the most frequent PIMs/PPOs in Figure 3.

The median number of AGS PIMs was two (IQR = 1–4; maximum 20). Eighteen per cent had no AGS PIMs, 18% one, 17% two, 14% three, 10% four, 7% had five, and 16% six or more (Table 2). For diuretics, tricyclic antidepressants (TCAs), selective serotonin reuptake inhibitors (SSRIs), or serotonin–norepinephrine reuptake inhibitors (SNRIs), for which there was the risk of inappropriate antidiuretic hormone secretion or hyponatremia, 49.9% of patients had at least one such medication. For peripheral α-1 blockers, which were to be avoided as antihypertensives, 43.7% had a prescription. Other frequent AGS PIMs included combinations of ≥ 3 central nervous system (CNS)-active drugs (17.2%), benzodiazepine receptor agonists (hypnotic Z-drugs) (16.6%), and proton pump inhibitors for > 8 weeks (12.6%) (Figure 3, Appendix A).

Patients had a median of three STOPP PIMs (IQR = 2–5, maximum 21), 11% had none, 13% one; 16% two; 16% three, 14% four, 10% five, and 21% six or more (Table 2). The most frequent PIMs were: 54.9% of patients took vasodilators although they had persistent postural hypotension, 48.4% were prescribed duplicates within drug classes such as nonsteroidal anti-inflammatory drugs (NSAIDs), SSRIs, or loop diuretics, 31.1% took regular opioids without laxatives, 20.9% took medications that increase bleeding risk or peptic ulcers (NSAIDs, vitamin K antagonists, direct thrombin or factor Xa inhibitors, or antiplatelet agents), and three of these medication groups which increase fall risk: (20.0% strong opioids, 16.6% hypnotic Z-drugs, 10.5% benzodiazepines, and 6.9% neuroleptics) (Figure 3, Appendix A).

Of the 27 START criteria that were assessed, the median was one PPO (IQR = 0–1, maximum 12), 49% of patients had none, 38% had one, 9% two, and 2% three (Table 2). The most frequent PPO was in 31.1% of patients, who were prescribed regular opioids without a laxative, 6.9% had osteoporosis but were not prescribed bone anti-resorptives, 5.7% had heart failure or coronary artery disease but were not prescribed angiotensin converting enzyme inhibitors (ACE) (Figure 3, Appendix A). It is encouraging that 49% had no PPOs. For the START criteria for which patients were receiving a needed medication, the median was one prescribed PPO (IQR = 0–2, maximum 12), 43% of patients had none, 31% had one, 14% two, 7% three, and 4% had four (Table 2). The principal corrected PPO was in 37.9% of patients, who were prescribed antihypertensive medications for high blood pressure (Figure 3, Appendix A).

### 3.2. Association of Medications with Health Outcomes

Possible associations between PIMs with readmission or mortality within six months of discharge were modeled by logistic regression (Table 3).

Male gender, age, comorbidities (as listed in Figure 2b), medications, and PIMs/PPOs all increased the odds of both outcomes measured by their crude odds ratios (all *p* < 0.001). The AGS and STOPP PIM totals showed moderate agreement with each other with a Pearson correlation coefficient of 0.7051 (95% CI 0.7016–0.7085, *p* < 0.001).

In multivariable models controlling for sex, age, and comorbidities, the number of medications, PIMs, PPOs, and needed PPO prescriptions all had significant effects on both readmission and mortality within six months of discharge. The adjusted odds of readmission within six months of discharge were increased by the number of medications (OR = 1.09, 95% CI 1.09–1.09, *p* < 0.001), AGS PIMs (OR = 1.15, 95% CI 1.14–1.16, *p* < 0.001), STOPP PIMs (OR = 1.15, 95% CI 1.14–1.15 *p* < 0.001), START PPOs (OR = 1.04, 95% CI 1.02–1.06, *p* < 0.001), and needed START prescriptions, which were prescribed for the patients (OR = 1.16, 95% CI 1.14–1.17, *p* < 0.001) (Table 3).

The odds of mortality within six months were increased minimally by the number of medications (OR = 1.02, 95% CI 1.01–1.02, *p* <0.001), but increased by the number of STOPP PIMS (OR = 1.07, 95% CI 1.06–1.08; *p* < 0.001), AGS PIMs (OR = 1.11, 95% CI 1.10–1.12, *p* < 0.001), START PPOs (OR = 1.31, 95% CI 1.27–1.34, *p* < 0.001), and minimally decreased by receiving needed START prescriptions (OR = 0.97, 95% CI 0.94–0.99, *p* < 0.0035) (Table 3). The impact of START PPOs (not being prescribed needed medications) on mortality indicates a likely higher prevalence of comorbidities/health issues in patients for whom the START criteria identified important needs.

Association rule mining analysis (ARM) was used to identify which individual or groups of PIMs/PPO medications were correlated with rehospitalisation or death within six months of discharge (Table 4 and Appendix A).

Four groups of STOPP PIMs had lift values with a > 3.5 times higher likelihood of mortality compared to if the data were randomly associated: (1) a lift value of 8.77 higher likelihood of mortality was associated with the combination of STOPP sets A3 (duplicate drugs from the same class), K2 (neuroleptics), and L1 (strong opioids), (2) a lift value of 8.67 higher likelihood of mortality was associated with the combination of STOPP sets K2 and L1, (3) a lift value of 3.95 higher likelihood of mortality was associated with the combination of STOPP sets A3 and K2, and (4) a lift value of 3.69 higher likelihood of mortality was associated with the STOPP set K2 (Table 4).

Two groups of STOPP PIMs with lift values were associated with ≥ twice the risk of readmission within six months: (1) a lift value of 2.36 higher likelihood of readmission was associated with the combination of STOPP sets A3, K2, and L1, (2) a lift value of 2.28 higher likelihood of readmission was associated with the combination of STOPP sets K2 and L1 (Table 4).

Three groups of AGS PIMs had lift values with a more than three times higher likelihood of mortality compared to if the data were randomly associated: (1) a lift value of 4.96 higher likelihood of mortality was associated with the combination of AGS sets 2E2 (neuroleptics and antipsychotics, except if prescribed for schizophrenia or bipolar disorder), 4D (antipsychotics, diuretics, SSRIs, SNRIs, and TCAs) and 5E (≥ 3 CNS-active drugs), (2) a lift value of 3.91 higher likelihood of mortality was associated with the combination of two AGS sets 2E2 and 4D, and (3) a lift value of 3.11 higher likelihood of mortality was associated with the combination of AGS sets 2E4 (benzodiazepines) and 5B (opioids with benzodiazepines) (Table 4).

Two groups of AGS PIMs with lift values ≥ 2 had more than twice the risk of readmission within six months: (1) a lift value of 2.24 higher likelihood of readmission was associated with the combination of AGS sets 2D1 (peripheral α-1 blockers) and 2G1 (metoclopramide) and (2) a lift value of 2.18 higher likelihood of readmission was associated with the combination of AGS sets 2G1 and 4D (Table 4).

Two groups of START PPO cardiovascular medications had high lift values for mortality: (1) the omission of an antihypertensive for hypertension (STARTA4) provided a lift = 3.56 for mortality and (2) the omission of ACE inhibitors for patients with heart failure/coronary artery disease (STARTA6) provided a lift = 2.38 for mortality. Two groups of PPO START cardiovascular medications had lower lift values for readmission: (1) the omission of beta blockers for patients with stable systolic heart failure (STARTA8) provided a lift = 1.46, and (2) for diabetic patients with renal disease the omission of ACE inhibitors or angiotensin II receptor antagonists (ARBs) (STARTF) provided a lift = 1.64 (Table 4).

The START PPOs correctly prescribed to patients were: (1) the combination of a statin with coronary, cerebral, or peripheral vascular disease (START A5) and an ACE inhibitor with systolic heart failure or coronary heart diseases (START A6), which had a small lift of 1.043 against mortality and (2) the combination of START A3 (antiplatelet therapy with coronary, cerebral, or peripheral vascular disease) (START A5 and A6) which had a similar small lift of 1.042 against mortality (Appendix A).

## 4. Discussion

This study identified high levels of PIMs and PPOs and statistically significant correlations with rehospitalisation and mortality within six months of discharge. The key solutions are to prevent inappropriate prescribing at the source before patients are prescribed such medications and prompt and effective deprescribing if they are prescribed. Systematic reviews describe low evidence for the effectiveness of deprescribing interventions. However, individual studies which have a more intensive deprescribing dialogue between pharmacists, hospitals, and family physicians were more effective. The barriers to deprescribing have been clearly described and what is missing are evidence-based studies for three interventions: detailed studies involving patients in decision making, improvements in the usability and curation of adverse drug event databases, and strong inducements at the executive level of health systems focused on avoiding PIMs, PPOs, and ADRs.

### 4.1. High Levels of Inappropriate Prescribing

In this study, medication appropriateness was evaluated for 82,935 patients ≥ 65 years who were hospitalised within a recent five-year period, accounting for over 50% of this demographic in Calgary, Canada. In this cohort, 82.0% were assessed as having ≥ 1 AGS PIMs and 89% had ≥ 1 STOPP PIMs, which is considerably higher than the literature values of weighted averages of 58.8% and 46.7%, respectively [7]. Higher levels of “potentially inappropriate prescribing” were found in substantial percentages of patients: 16% had three, 14% four, 10% five, and 21% six or more STOPP PIMs and 14% had three, 10% four, 7% five, and 16% six or more AGS PIMs. These percentages may be partly due to the median number of prescriptions increasing from four at admission to nine at discharge, and the AGS PIMs also included medications with cautions.

### 4.2. Relationship of Inappropriate Prescribing and Health Outcomes

Multivariable models controlled for sex, age, and comorbidities (as listed in Figure 2b), and the number of medications, PIMs, PPOs, and needed PPO prescriptions all had significant effects on both readmission and mortality within six months of discharge (Table 3). In the ARM analysis, drugs that correlated with both mortality and readmission within six months of discharge were strong opioids, duplicate drugs from the same class, neuroleptics and antipsychotics, diuretics, SSRIs, SNRIs, and TCAs. Other drugs that correlated with the outcomes were ≥ 3 CNS-active drugs, benzodiazepines, and opioids with benzodiazepines with mortality, and peripheral α-1 blockers and metoclopramide with readmission. Thus, CNS and cardiovascular drugs predominated in these correlations and should be a major focus for prescribers (Table 4).

### 4.3. Interventions to Improve Polypharmacy

The interventions usually used to assess polypharmacy and reduce the harms ascribed to polypharmacy are chart and medication reviews by physicians and pharmacists. A systematic review identified 32 studies of complex interventions by physicians and pharmacists to improve polypharmacy for older people. The studies employed a variety of designs (18 randomised trials, 10 cluster randomised trials, two non-randomised trials, and two controlled before–after studies). The risk of bias was high and/or unclear on the Cochrane risk of bias tool for several domains and the certainty of evidence on the GRADE tool was low to very low. It was uncertain whether the interventions improved the appropriateness of medications (mean difference −4.76; 95% CI −9.20, −0.33), reduced PIMs (risk ratio 0.79; 95% CI 0.61, 1.02) or PPOs (standard mean difference −0.81; 95% CI −0.98, −0.64), reduced hospitalisation rates, or improved the quality of life [24].

However, some individual studies, which describe more intense interaction between deprescribers, are encouraging. An RCT in Ireland of 737 patients used pharmacists and software decision support and found physicians accepted 55% of discontinuation recommendations [25]. An RCT in Québec found that, at six months, 43% in the intervention and 12% in the control were no longer filling PIMs [26].

The review [24] did not assess discussions with patients or their families about their medication appropriateness and the key component missing in most studies is a description of detailed interactions and discussions between deprescribers and patients [24]. In a study of 400 patients in Sweden, pharmacists had detailed discussions with patients and their families and identified PIMs and PPOs. After twelve months, PIMs and PPOs were significantly reduced and drug-related admissions were reduced by 80% and emergency visits by 47% [27,28]. In a study in Japan, pharmacists identified 651 PIMs in 822 patients ≥ 65 and physicians discontinued 45% of the recommended medications [29].

### 4.4. Barriers to Deprescribing

A systematic review identified 116 studies of deprescribing for older patients (average 74 years of age) but the RCTs overall showed no effect on mortality (OR 0.82, 95% CI 0.61–1.11) although patient-specific interventions did (OR 0.62, 95% CI 0.43–0.88) [30]. A qualitative systematic review of 21 studies identified multiple barriers to deprescribing with older primary care patients, which need to be overcome: (1) poor awareness by physicians of their overprescribing, (2) deprescribing involves patient resistance to change, poor acceptance of alternatives, increased workload, and the need to cope with patients’ withdrawal symptoms, (3) high prescribers downplay risks of harms, (4) physicians have deficiencies in knowledge and confidence about clear-cut indications for medications, recognising ADRs, and balancing medication benefits and harms, especially for older multimorbid polypharmacy patients, and (5) inadequate time to review and discontinue medications [31]. It will take substantial effort to overcome these barriers in the individual physician–patient relationship. A systematic review of RCTs to encourage family physicians to increase rational screening testing for chronic diseases showed many educational initiatives have limited effectiveness [32]. Similarly, it will be necessary to test multiple components in deprescribing initiatives to overcome the above barriers, and require strong and frequent institutional support and incentives.

### 4.5. Oversight at the Level of Healthcare Systems

A WHO review identified a lack of information about coordination and integration at the organisational level and the oversight needed to optimise medications and care for populations of senior patients [33]. To identify PIMs/PPOs and prioritise patients requiring extra medication assessments and care, including system-level applications, will require an active organisation with considerable authority.

ADR reporting systems need to be much more patient- and professional-friendly. A systematic review identified 108 ADR reporting systems, including 1782 unique data fields, and their complexity limited data aggregation and evaluation [34]. The U.S. Department of Veterans Affairs database helpfully provides ADR information by 10-year age groups (but data are nearly all for males) [6]. The U.S Food and Drug Administration maintains a public Adverse Event Reporting System, which due to its large size and complexity, makes data mining difficult [35,36] and the latest analysis only summarizes reports from 2006–2014 [37]. On a global scale, systematic reviews of ADRs are rare [38].

### 4.6. Limitations

This is a retrospective database study and all relationships are correlational. The ethics approval required that we did not have access to any charts and all data were anonymised before we received them. Thus, we were not able to review charts or any handwritten progress notes. The PIMs and PPOs that we could not assess are noted above in the methods section. Because we did not have access to the charts, we could not assess the reasons for additional medications during admission (some of which were PIMs) and why prescriptions were provided to correct some PPOs and not others.

There was no specific category for ADRs in the hospitals’ electronic medical records. Our estimate of confidence in the rate we derived was very low and we chose not to report it rather than report data for which the quality of reporting was unknown.

Cox proportional hazard models were originally performed to more accurately characterise survival, however, the non-proportional hazard assumption could not be met by the correction of time–interaction terms and we used logistic regression instead.

## 5. Conclusions

The study population was similar in the five-year age groups to the same age populations of Alberta, Canada and the US and could be generalised to those jurisdictions on this limited basis (Appendix A). The average age of the sample was 75 years and they took a median of four (IQR 2–7) medications on admission and nine (IQR 5–13) at discharge. The most frequent medication class at discharge was statins (prescribed to 44.7% of patients), followed by alpha-1 blockers (43.7%), proton pump inhibitors (42.1%), opioids (39.8%), and beta blockers (36.9%) (Figure 2a).

Their comorbidities, in order of frequency, were hypertension, diabetes, cardiovascular disease, arthritis, dementia or cognitive decline, atrial fibrillation, falls, fractures, osteoporosis, pneumonia, heart failure, COPD, urinary tract infection, pain, renal failure, coronary artery disease, myocardial infarction, peptic ulcer, stroke, and chronic renal disease. This study shows that the STOPP/START and AGS PIMS, PPOs, and PPOs needing prescribing are key medications related to these comorbidities and to adverse outcomes of rehospitalisation and death within six months of admission.

Systematic reviews show that most studies of interventions using pharmacists to improve medication appropriateness were at a high risk of bias and their effectiveness was uncertain [24] but that more intensive cooperation with physicians is effective [25,26,29]. RCTs using intensive interventions and detailed and thorough discussions, in which patients and their families are comprehensively informed and involved in all decisions, are effective and more studies focused on patients’ observations and preferences are needed [27,28].

A systematic review of 116 studies of encouraging physicians to deprescribe for older patients showed no effect on mortality but those studies which involved patients were effective [30]. More studies of interventions to involve patients and their families to assess the benefits and risks of deprescribing individual medications and individualising care for each patient are needed. The barriers for physicians to deprescribe are poor awareness of potential ADRs, especially in high prescribers, and inadequate knowledge of how to decrease polypharmacy safely in patients with multimorbidity and meet their needs to ameliorate their symptoms [31,32]. The WHO has emphasised that health organisations need to increase the monitoring of the care provided to the older patients in their care. This needs to include regularly reviewing the medications of all seniors in their care by applying AGS and STOPP/START criteria, and providing educational sessions and clinics in which physicians and pharmacists discuss with patients and their families how to optimise the medications of individual seniors. Medications are a complex problem and enough time, expertise, and management oversight need to be devoted to their appropriate solution.

The AGS and STOPP/START are the most widely used criteria and findings using these criteria can be generalised to other countries within the limits of the varying availability and use of medications in different countries and the need to regularly update the criteria or supplement them as new medications are added and older ones deleted from use.

## Figures and Tables

**Figure 1 geriatrics-05-00037-f001:**
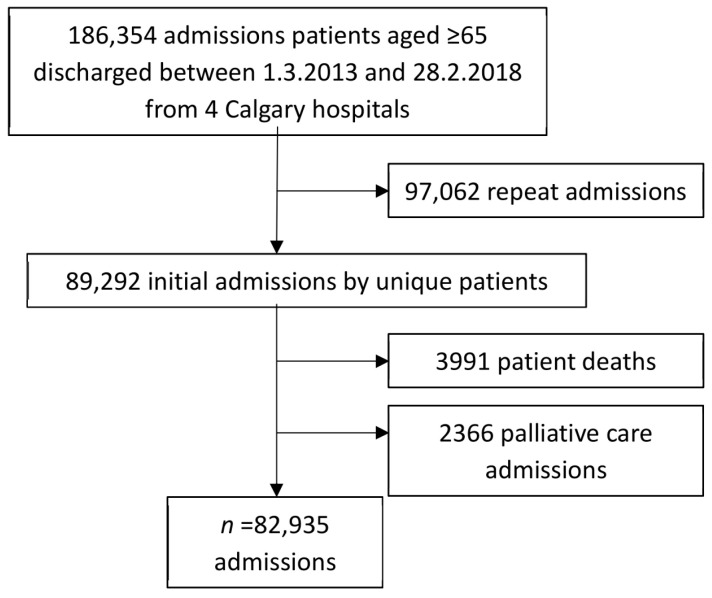
The admission of 186,354 patients aged ≥ 65 to four Calgary hospitals.

**Figure 2 geriatrics-05-00037-f002:**
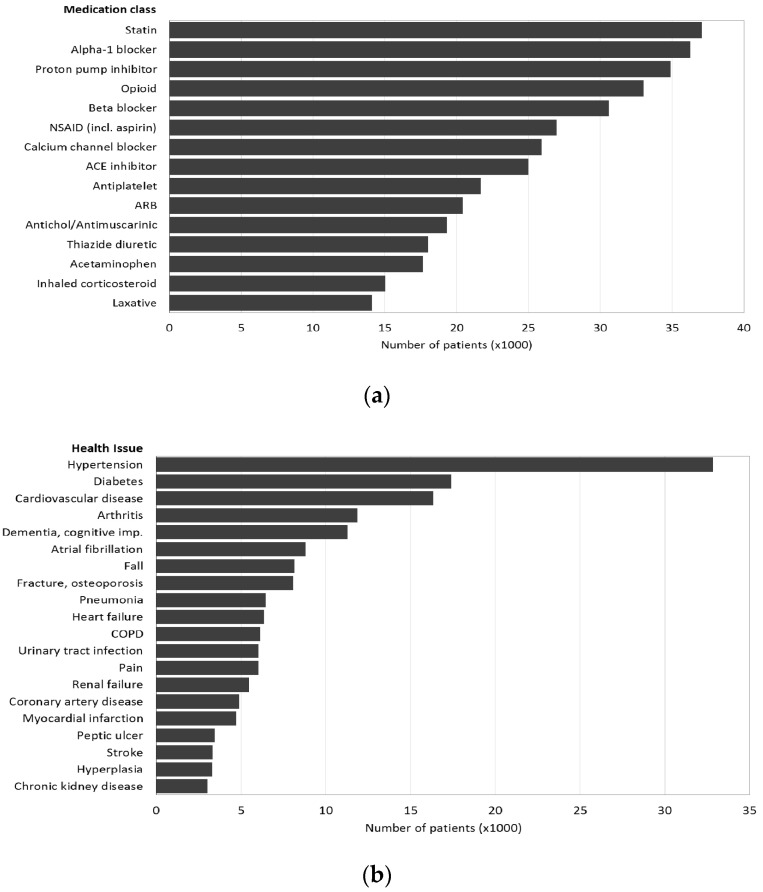
Notes: ARB = angiotensin receptor blocker; COPD = chronic obstructive lung disease. (**a**) Medication class; (**b**) health issues.

**Figure 3 geriatrics-05-00037-f003:**
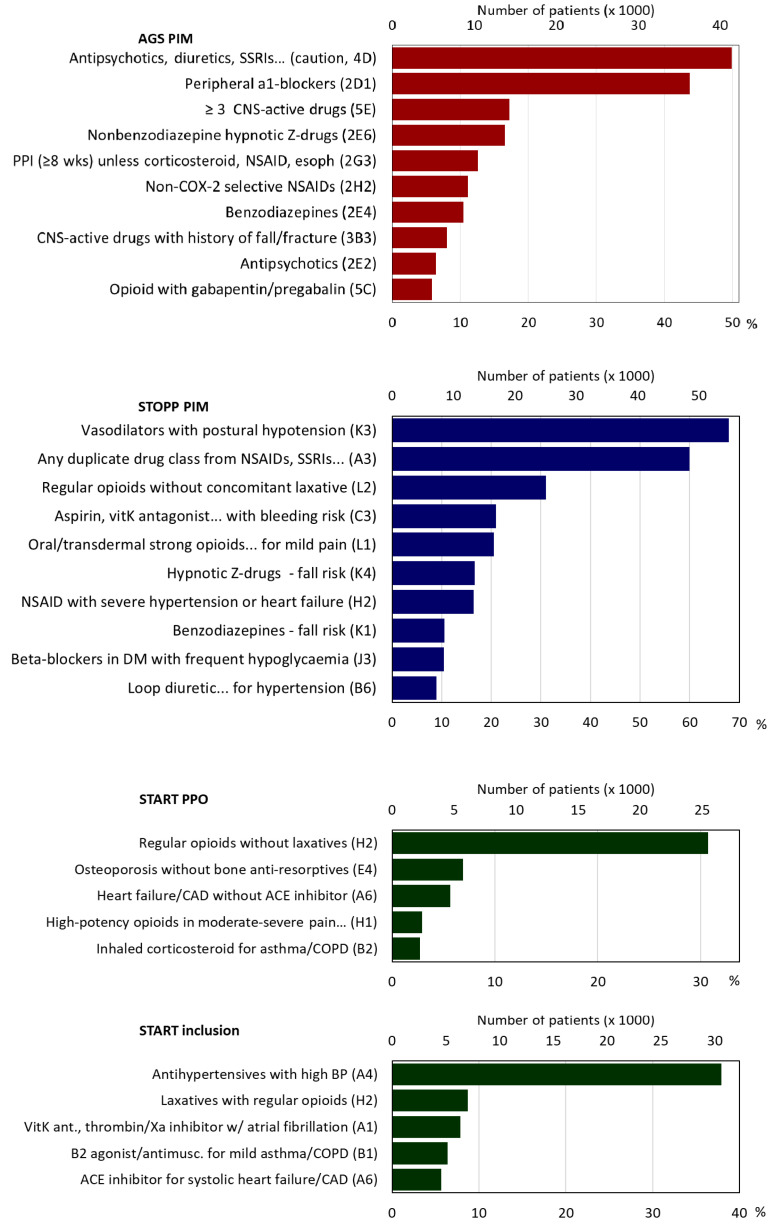
AGS PIMs, STOPP PIMs, START PPOs, START prescriptions. Note: PIM = potentially inappropriate medication; AGS = American Geriatric Society; PPO = potential prescribing omission; SSRI = selective serotonin reuptake inhibitor; CNS = central nervous system; Z-drugs = hypnotic drugs beginning with the letter Z, e.g., Zaleplon, Zolpidem, Zopiclone; NSAID = nonsteroidal anti-inflammatory drugs; CAD = coronary artery disease; ACE inhibitor = angiotensin-converting enzyme inhibitors; BP = blood pressure.

**Table 1 geriatrics-05-00037-t001:** Summary characteristics for older patients admitted to Calgary hospitals, 2013–2018.

	Both Genders	Female	Male
Number of patients (%)	82,935 (100)	41,866 (50.5)	41,069 (49.5)
Age group			
65–69	23,675 (28.5)	10,814 (13.0)	12,861 (15.5)
70–74	17,336 (20.9)	8174 (9.9)	9162 (11.0)
75–79	14,728 (17.8)	7422 (8.9)	7306 (8.8)
80–84	12,962 (15.6)	6789 (8.2)	6173 (7.4)
85–89	9088 (11.0)	5219 (6.3)	3869 (4.7)
90+	5146 (6.2)	3448 (4.2)	1698 (2.0)
Overall age			
Median	75	76	74
IQR	69–82	70–84	69–81
Medicines upon admission			
Median	4	4	4
IQR	2–7	2–7	2–7
Maximum	28	26	28
Medicines upon discharge			
Median	9	9	9
IQR	5–13	5–13	5–13
Maximum	63	60	63
Outcome frequencies, 6M post discharge (%)			
Readmission	18,777 (22.6)	9018 (10.9)	9759 (11.8)
Mortality	4606 (5.6)	2214 (2.7)	2392 (2.9)

**Table 2 geriatrics-05-00037-t002:** Frequency distributions of STOPP PIMs, AGS PIMs, START PPOs and START prescriptions (82,935 admissions).

No. of Violations	STOPP PIMs (%)		START PPOs (%)		START Inclusions (%)		AGS PIMs (%)	
0	8680	(10.5)	40,513	(48.8)	35,958	(43.4)	14,925	(18.0)
1	10,413	(12.6)	31,609	(38.1)	25,574	(30.8)	15,080	(18.2)
2	13,570	(16.4)	7772	(9.4)	11,309	(13.6)	13,917	(16.8)
3	12,987	(15.7)	2015	(2.4)	5405	(6.5)	11,376	(13.7)
4	11,402	(13.7)	666	(0.8)	2882	(3.5)	8279	(10.0)
5	8655	(10.4)	228	(0.3)	1122	(1.4)	5920	(7.1)
6	6277	(7.6)	87	(0.1)	428	(0.5)	4225	(5.1)
7	4306	(5.2)	26	(0.0)	179	(0.2)	3082	(3.7)
8	2787	(3.4)	16	(0.0)	52	(0.1)	2245	(2.7)
9	1724	(2.1)	2	(0.0)	18	(0.0)	1580	(1.9)
10	1002	(1.2)	0	(0.0)	5	(0.0)	974	(1.2)
> 10	1132	(1.4)	1	(0.0)	3	(0.0)	1331	(1.6)
Median	3		1		1		2	
IQR	2–5		0–1		0–2		1–4	
Maximum	21		12		12		20	
Note: IQR = interquartile range								

**Table 3 geriatrics-05-00037-t003:** Crude and adjusted odds ratios for AGS, STOPP, or START criteria violations in logistic regression models for patient hospital readmission or mortality within 6 months of discharge ^a^.

Independent Variable	Readmission (95%CI)	Mortality (95%CI)
Crude OR		
Gender (M)	1.14 (1.10–1.17)	1.11 (1.04–1.18)
Age	1.02 (1.02–1.03)	1.07 (1.07–1.08)
Comorbidities	1.12 (1.11–1.13)	1.21 (1.20–1.23)
Medications	1.09 (1.09–1.10)	1.03 (1.03–1.04)
Beers PIMs	1.13 (1.13–1.14)	1.11 (1.10–1.12)
STOPP PIMs	1.16 (1.16–1.17)	1.12 (1.11–1.14)
START PPOs	1.10 (1.08–1.12)	1.49 (1.45–1.53)
START inclusions	1.22 (1.21–1.24)	1.12 (1.10–1.15)
Adjusted OR ^b^		
Medications	1.09 (1.09–1.09)	1.02 (1.01–1.02)
AGS PIMs	1.14 (1.13–1.14)	1.11 (1.10–1.12)
STOPP PIMs	1.15 (1.14–1.15)	1.07 (1.06–1.08)
START PPOs	1.04 (1.02–1.06)	1.31 (1.27–1.34)
START inclusions	1.16 (1.14–1.17)	0.97 (0.94–0.99, *p* = 0.0035)

^a^ all *p* < 0.001 unless otherwise indicated. ^b^ adjusted by patient age, gender, and comorbidities.

**Table 4 geriatrics-05-00037-t004:** Adverse combinations of selected medications identified by association rule mining.

Medication Combinations	Increased Likelihood * of Readmission within 6 Months
STOPP: duplicate drug class + neuroleptics + oral/transdermal strong opioids	2.36
STOPP: neuroleptics + oral/transdermal strong opioids	2.28
AGS: peripheral alpha 1 blocker + metoclopramide	2.24
AGS: antipsychotics + diuretics + ≥ CNS active drugs (antidepressants, benzodiazepines, or opioids) + metoclopramide	2.18
START for diabetic patients with renal disease the omission of ACE inhibitors or angiotensin II receptor antagonists (ARBs)	1.64
START: omission of beta blockers for patients with stable systolic heart failure	1.46
STOPP: duplicate drug class + neuroleptics + oral/transdermal strong opioids	8.77
STOPP: neuroleptics + oral/transdermal strong opioids	8.67
STOPP: duplicate drug class + neuroleptics	3.95
STOPP: neuroleptics	3.69
AGS: antipsychotics + diuretics + ≥ three CNS active drugs (antidepressants, benzodiazepines, or opioids)	4.96
AGS: antipsychotics + ≥ CNS active drugs (antidepressants, benzodiazepines, or opioids)	3.91
AGS: benzodiazepines + opioids	3.11
START: hypertension but no HTN Rx	3.56
START: heart failure or CAD without ACE inhibitor	2.38

Notes: * ARM compares groups of two or more medications to a random section of medications from the database of all medications taken by all patients in the study and reports the likelihood of a selected outcome, such as rehospitalisation or mortality compared to the random computer-selected medications; CNS = central nervous system; ACE inhibitor = angiotensin-converting enzyme inhibitor; HTN = hypertension; CAD = coronary artery disease.

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
