# Peer review of "Potentially Inappropriate Prescribing and Potential Prescribing Omissions in 82,935 Older Hospitalised Adults: Association with Hospital Readmission and Mortality within Six Months"

_geriatrics, 2020, doi:10.3390/geriatrics5020037_

Round 1
Reviewer 1 Report
Well written paper and very relevant in the current climate of needing to improve aged care quality.
a minor typo in line 58 where the word 'in' is repeated
Line 85 - AGS is written out in full in the abstract and later in the method but should have been introduced here in the background in the first instance of the main paper.
Review the formats within the references - the fonts are a mix of several different ones
Author Response
Response to Reviewer number one
Reviewer’s comments and suggestions |
Authors’ response |
Well written paper and very relevant in the current climate of needing to improve aged care quality. |
Many thanks for your encouraging comment |
a minor typo in line 58 where the word 'in' is repeated |
corrected |
Line 85 - AGS is written out in full in the abstract and later in the method but should have been introduced here in the background in the first instance of the main paper. |
corrected |
Review the formats within the references - the fonts are a mix of several different ones |
Thank you. Corrected. |
Reviewer 2 Report
Dear Author(s),
This is a great study for the readers of the Geriatrics. I would suggest the following adjustments to the followings:
Background and Intro:
It needs some modifications. There are long and confusing paragraphs, exp: Lines 45-48: needs rephrasing, lines 69-75.
Definition of 5 and more for polypharmacy is applicable to adults not pediatrics, please add " adults"
Method:
This section needs some major modifications. There are several subtitles could be merged and summarized. Several lines to be removed by using a simple citation. ( exp lines: 121- 127, 174-188)
Under variable( 2.3) : This section needs some clarification for the covariates and how authors were able to identified them without accessing to EHR and chart review ( elaborate 2.4).-Age bracket, every 5 years
-Medication classification: citation?
There are several subtitle- subsections that can be merged and summarized into smaller paragraphs. ( 2.5. Bias: is a confusing subtitle, can be places under measurement …, the rest of this section to be placed under limitations in discussion.) Also, Section 2.7: part of this paragraph must be placed in data collection and summarized.
Please also avoid some repetitive information across the paper.
Results:
This section needs to start with the sample size again: 82,935. Please also summarize this section and remove all the subtitles. There could be referral to the tables by number which is easier for the readers to review and compare.
Figure 2 and 3 could go to appendix
Table 3, can you provide separate line for each comorbidity?
Discussion:
It is also suggested to remove the subtitles and summarized the long paragraphs, or remove the repetitive information. Limitation needs to be elaborated and provide detailed information.
The section of generalizability can also be titled ( with modification and restructuring ) as conclusion.
Author Response
Replies to Reviewer number two
Reviewer comments and suggestions |
Authors’ replies |
Dear Author(s), This is a great study for the readers of the Geriatrics. I would suggest the following adjustments to the followings:
|
Thank you very much for your very helpful suggestions. Your ideas about restructuring and simplifying the presentation and removing unnecessary headings are very much appreciated. |
Background and Intro: It needs some modifications. There are long and confusing paragraphs, exp: Lines 45-48: |
This has been deleted: Polypharmacy involving inappropriate prescribing can take many forms: safer medications could be prescribed, beneficial medications are not prescribed, medications are prescribed in inappropriate numbers, doses or durations, or significant medication-medication, medication-medication-gene or medication-disease interactions occur so that medications cause harm. We did a literature search for a systematic review on multimorbidity and polypharmacy, which is the heart of the polypharmacy problem and we replaced the above paragraph with this: Family physicians care for patients with multimorbidity in up to 80% of their consultations and geriatricians in nearly all of their consultations. These patients take multiple medications and a systematic review concluded that to avoid interactions between the vast potential heterogeneity of illnesses and polypharmacy requires ascertaining patient goals and preferences, shared goal setting and decision making, and individualised care with excellent continuity to optimize benefits over harms.
[Reference is: Muth, C.; Blom, J.W.; Smith, S.M.; Johnell, K.; Gonzalez-Gonzalez, A.I.; Nguyen, T.S; Brueckle, M.-S.; Cesari, M.; Tinetti, M.E; Valderas, J.M. Evidence supporting the best clinical management of patients with multimorbidity and polypharmacy: a systematic guideline review and expert consensus. J Intern Med 2019;285: 272–288]. |
needs rephrasing lines 69-75. |
Revised, with these deletions:
Adverse drug events are more frequent among seniors. Another concern is whether adverse drug events are more frequent among older seniors. A review of adverse drug reactions in The US Veterans Affairs database 2009-2016 identified a 15% rate of adverse drug reactions of 15% for those 60-69, 13% for 70-79, 11% for 80-89 and 9% for those 90 and older and with The rates of 5% of ADRs were rated as severe. ADRs of 5% for those 60-69 and rising to 6% for those 90 and older. Adverse drug events were related mostly to cardiovascular and diabetes medications. The rank order of individual medications involving ADRs for those 90 and older was: 1. warfarin, 2. lisinopril, 3. Sulfamethoxazole-Trimethoprim, 4. aspirin, 5. simvastatin, 6. clopidogel, 7. furosemide, 8. metoprolol, 9. heparin and 10. Glipizide, with similar rankings for those 60-90. Again, individual medications related to the cardiovascular system and diabetes dominated [5].
|
Definition of 5 and more for polypharmacy is applicable to adults not pediatrics, please add " adults" |
“adults” was added |
Method: This section needs some major modifications. There are several subtitles could be merged and summarized. |
These headings were deleted: 2.2. Participants and Setting 2.5. Bias 2.6. Study Size 2.7. Statistical Methods This text was deleted: “Potential Prescribing Omissions” (PPOs) were assessed using the Screening Tool to Alert to Right Treatment (START) [7]. Both sets of criteria were developed by panels of geriatricians and geriatric psychiatrists using comprehensive literature searches and Delphi methods. The AGS criteria were updated in 2019, and the STOPP/START criteria in 2015 and the next update will be in 2020. AGS PIMs are listed in a series of tables: medications to be avoided or to be avoided because they result in drug-drug or drug-syndrome interactions; or to be used with caution; or at reduced dosage according to renal function levels; or avoided because of strong anticholinergic properties. The STOPP criteria list 80 STOPP PIMs and 34 START PPOs on an anatomical/therapeutic basis.
|
Several lines to be removed by using a simple citation. (exp lines: 121- 127), |
These lines have been deleted and replaced with a citation number |
Lines need to be removed and replaced by a citation (174-188)
|
These lines have been deleted and replaced with a citation number |
Under variable (2.3): This section needs some clarification for the covariates and how authors were able to identified them without accessing to EHR and chart review (elaborate 2.4).- |
This sentence has been added: Admission and discharge diagnoses and comorbidities were obtained from admission and discharge summaries.
|
Age bracket, every 5 years
|
We decided that a decade is a long period for the ≥ 65s during which to assess for change, because some individuals progress in their illnesses and may change more rapidly over much shorter periods. This sentence has been added: Five-year age groups were chosen to identify ages during which adverse changes occurred. |
Medication classification: citation? |
This has been added: The medications were classified by anatomical then therapeutic classifications (A/T) by both STOPP/START and the American Geriatric Society Beers criteria and their detailed listing of individual medications under their A/T classifications are provided in Tables S2, S3 and S4. |
There are several subtitle - subsections that can be merged and summarized into smaller paragraphs |
Subtitle sections have been removed an shorter paragraphs merged |
( 2.5. Bias: is a confusing subtitle, can be places under measurement …, the rest of this section to be placed under limitations in discussion.) |
Agreed, Bias has been removed as a heading. Within that section this sentence has been moved to limitations:
Cox proportional hazard models were originally performed to more accurately characterize survival, however the non-proportional hazard assumption could not be met by correction of time-interaction terms and we used logistic regression instead. |
Also, Section 2.7: part of this paragraph must be placed in data collection and summarized. |
This has been placed in data collection and summarised |
Please also avoid some repetitive information across the paper. |
We have reread the entire paper and removed repetitions. |
Results: This section needs to start with the sample size again: 82,935. |
He sample size of 82,935 has been added |
Please also summarize this section and remove all the subtitles. There could be referral to the tables by number which is easier for the readers to review and compare. |
These headings have been removed: 3.2. Main Results: Frequencies of Medications, PIMs and PPOs 3.3.1. Multiple Logistic Regression Analysis 3.3.2. |
Figure 2 and 3 could go to appendix. |
I hope you don’t mind if we retain them in the body of the paper. Readers will want to ask themselves several questions to evaluate if the study is relevant to their patients before reading further. What were the most frequent illnesses in order of frequency? What were the most frequent medications in order of frequency? (because this is a study which focuses on polypharmacy).
I have found that after an interval of a couple of years the link to the supplemental material for an article may no longer works. Even our inter-library loan staff cannot find the link. |
Table 3, can you provide separate line for each comorbidity?
|
A great idea. Regretfully the statistician on the project got a promotion to data analyst in the Alberta Laboratory services and the Department of Pathology and became snowed under with Covid19 work. He said he had no time to do this. He coded the data in RStudio which the rest of us are not able to do. |
Discussion: It is also suggested to remove the subtitles and summarized the long paragraphs, or remove the repetitive information. |
This heading was removed and repetitive information deleted: 4.6. and Curation and Integration of ADR Reporting Systems
|
Limitation needs to be elaborated and provide detailed information.
|
The following sentences were added to limitations:
Because we did not have access to the charts we could not assess the reasons for additional medications during admissions (some of which were PIMs) and why prescriptions were provided to correct some PPOs and not others.
There was no specific category for ADRs in the hospitals’ electronic medical records. Our estimate of confidence in the rate we derived was very low and we chose not to report it rather than report data for which the quality of reporting is unknown.
Cox proportional hazard models were originally performed to more accurately characterize survival, however the non-proportional hazard assumption could not be met by correction of time-interaction terms and we used logistic regression instead.
|
The section of generalizability can also be titled (with modification and restructuring) as conclusion. |
The heading 4.8. Generalisability was deleted and replaced by Conclusions
A new conclusions section was written:
The study population was similar in 5-year age groups to the same age populations of Alberta, Canada and the US and could be generalised to those jurisdictions on this limited basis (Table S5). The average age of the sample was 75 years and they took a median of four (IQR 2-7) medications on admissions and 9 (IQR 5-13) at discharge. The most frequent medication class at discharge was statins (prescribed to 44.7% of patients), followed by alpha-1 blockers (43.7%), proton pump inhibitors (42.1%), opioids (39.8%) and beta-blockers (36.9%) (Figure 2a).
Their comorbidities in order of frequency were hypertension, diabetes, cardiovascular disease, arthritis, dementia or cognitive decline, atrial fibrillation, falls, fractures, osteoporosis, pneumonia, heart failure, COPD, urinary tract infection, pain, renal failure, coronary artery disease, myocardial infarction, peptic ulcer, stroke, and chronic renal disease.
The adjusted odds of readmission within six months of discharge were increased by numbers of medications in the following order: START PPOs (OR = 1.04, 95%CI 1.02-1.06, p <0.001); total number of medications (OR = 1.09, 95%CI 1.09-1.09, p <0.001); AGS PIMs (OR = 1.15, 95%CI 1.14-1.16, p <0.001); STOPP PIMs (OR = 1.15, 95%CI 1.14-1.15 p <0.001); and needed START prescriptions which were prescribed to the patients (OR = 1.16, 95%CI 1.14-1.17, p <0.001) (Table 3).
The odds of mortality within six months were increased by numbers of medications in the following order: total number of medications (OR = 1.02, 95%CI 1.01-1.02, p <0.001); STOPP PIMS (OR = 1.07, 95%CI 1.06-1.08; p <0.001); AGS PIMs (OR = 1.11, 95%CI 1.10-1.12, p <0.001); START PPOs (OR = 1.31, 95%CI 1.27-1.34, p <0.001). (Table 3). The largest impact on mortality was due to not prescribing needed START PPOs whereas receiving needed START prescriptions (OR = 0.97, 95%CI 0.94-0.99, p <0.0035) did decrease mortality minimally. Assessing START criteria identifies important needs and should be a key area for further investigation and prescribing.
The AGS and STOPP/START are the most widely used criteria and findings using these criteria can be generalised to other countries within the limits of the varying availability and use of medications in different countries and the need to regularly update the criteria or supplement them as new medications are added and older ones deleted from use.
|